# Towards Large Particle Size in Compound Feed: Using Expander Conditioning Prior to Pelleting Improves Pellet Quality and Growth Performance of Broilers

**DOI:** 10.3390/ani12192707

**Published:** 2022-10-08

**Authors:** Marco Antônio Ebbing, Nadia Yacoubi, Victor Naranjo, Werner Sitzmann, Karl Schedle, Martin Gierus

**Affiliations:** 1Institute of Animal Nutrition, Livestock Products, and Nutrition Physiology (TTE), Department of Agrobiotechnology, IFA-Tulln, University of Natural Resources and Life Sciences, 1190 Vienna, Austria; 2Evonik Operations GmbH, Rodenbacher Chaussee 4, D-63457 Hanau, Germany; 3Evonik Guatemala S.A., 18 calle 24-69, zona 10, Edificio Empresarial Zona Pradera, Torre 4, Oficina 810, Ciudad de Guatemala 01010, Guatemala; 4Amandus Kahl GmbH & Co. KG, Dieselstrasse 5, D-21465 Reinbek, Germany

**Keywords:** specific mechanical energy, processing, grinding, feed efficiency, AME_N_

## Abstract

**Simple Summary:**

The poultry meat industry is an important producer of high-quality protein. Broilers require well-balanced and processed diets, which should be environmentally friendly and produced in a sustainable way. The current study was designed to explore opportunities in the feed processing line; it aimed to improve resource savings and broiler performance by changing the particle size of the major ingredients in the diet using hammer or roller mills, and then, pelleting the compound diet using two different conditionings, with an expander or without, prior to pelleting. The hypothesis was that broilers perform better when fed diets processed using a combination of grinding the coarse particles with a roller mill and expanding them before pelleting. The results show that the processing line promotes intense secondary grinding, reducing the particle size. The use of a roller mill, coarse particles and an expander resulted in increased energy utilization, nutrient digestibility and feed conversion ratio of the broiler chickens.

**Abstract:**

During the processing of compound feed for broilers, several changes occur that affect the physical and probably the nutritional properties of pellets, influencing animal performance. The effects of mill type, particle size (PS) and expander conditioning prior to pelleting (E + P) were combined to generate pellets. A 2 × 3 × 2 factorial arrangement was designed with two mill types (a hammer mill (HM) or roller mill (RM)), three PSs (0.8, 1.2 or 1.6 mm) and two E + Ps (with or without expander processing prior to pelleting), with six replications of 12 unsexed Ross 308 broilers each. All the processing lines reduced the PS from mash to finished pellets via secondary grinding, by 2.35 times on average. However, RM grinding required less electric power (*p* < 0.001). The intended PS (0.8, 1.2 or 1.6 mm) did not affect this energy consumption. E + P and the PS interacted for the pellet durability index (PDI) (*p* = 0.006). The worst PDI in the pellets was observed when a PS of 1.6 mm without E + P was used. Only E + P positively affected starch (*p* < 0.001) and amino acids’ ileal apparent digestibility (*p* < 0.01). Organic matter (OM) (*p* = 0.02) and fat (*p* < 0.001) digestibility, as well as AME_N_ (*p* = 0.005) content, were influenced by the PS (main effect), whereas E + P and mill type interacted with these values (*p* < 0.005). Lower OM digestibility and AME_N_ content were observed when RM without E + P was used (*p* = 0.001). The feed conversion ratio (FCR) was enhanced and feed intake (FI) was improved with E + P. The combination of the RM mill, a 1.6 mm mean PS, and E + P improved FCR (three-way interaction, *p* = 0.019)), showing that for a higher PS, E + P is necessary for animal performance. Carcass yield was, on average, 80.1%. No effects on commercial cuts (breast, legs and wings) were observed. In contrast, abdominal fat was affected by mill type * PS (*p* = 0.012) and E + P * PS (*p* = 0.048) in a two-way interaction. The highest abdominal fat indicated an imbalance in the amino acid (AA)-to-AME_N_ ratio. Coarse PS promoted heavier gizzards (*p* = 0.02) but E + P tended to reduce them (*p* = 0.057). The processing steps improved pellet quality and feed efficiency associated with RM, coarse PS and E + P, highlighting the positive effects of E + P on abdominal fat and AME_N_ content, which should be adjusted to AA or reduced at formulation. However, these results are for an experimental processing plant and may not necessarily apply to larger plants, so the use of these data and methods should be considered as guidelines for replication at production sites.

## 1. Introduction

The current diet formulation of broiler compound feed includes large numbers of ingredients, including by-products from the food and feed industries. Many of these byproducts, prior to compound feed production, show a wide range of particle sizes resulting from the transformation steps of those by-products. Most by-products have lost their original grain or seed form due to the transformation steps; these include extraction meals, brans, or other ingredients, which are included in compound feeds with small particle sizes. Ingredients with small particle sizes reduce the average particle size of the diet [1]. To regulate the average particle size reduction in compound feeds for broilers, the cereal ingredient is the only intact whole grain; this can be used to manage the average particle size of the compound feed, and higher pellet quality and nutrient digestibility are expected [2,3]. In corn–soybean meal (SBM) diet formulations, the corn component may be ground, allowing some adjustment of the final particle size of the compound feed. In contrast, whole kernels of soybean no longer remain in the soybean meal, as full-fat soybeans are crushed or flaked, primarily for oil extraction, thereby reducing the particle size in this step [4]. 

The adjustment of the particle size of ingredients in compound feed is achieved by using hammer or roller mills, via single or combined grinding. The main transformation of particles in the milling processing of these ingredients is through physical change in their size and shape [5,6]. According to Koch [7], hammer mills produce spherical particles with polished surfaces; these are different from those produced in a roller mill, which are generally more rectangular or cubic rather than spherical [8,9]. Whatever the grinding technique used and shape of particles obtained, the particles generated after milling are submitted to further steps in the processing line, and this may contribute to further changes in particle size or particle form; this is mainly due to the influence of mechanical stress from the machines, as well as moisture, pressure and temperature. However, secondary grinding in the processing plant has been less quantified. The intensity of secondary grinding may influence the target particle size aimed for by the mill type, i.e., the hammer or roller mill. Therefore, the mill type alone does not have a large influence on the animal performance response when the final product has additionally been conditioned and pelleted. In fact, in one study, the intended particle size after grinding did not seem to be the same as that which the birds pecked when pelleted diets were provided [10]. However, until now, the influence of secondary grinding on particle size in pellets and on animal performance after consuming those pellets has been poorly quantified. 

Using pelleted compound feeds with different particle sizes for poultry has been well documented [11]. Ideally, pellets should have a low proportion of fine parts of feed [12]. In this way, the proportion of intact pellets available, as measured by the pellet durability index (PDI), should be high. Mash with small particle size has been reported as beneficial for producing pellets with a high PDI [7,11]. However, a coarse structure is required for broiler gizzard development and activity, with the pH dropping in the proventriculus, pepsin attachment and amino acid digestibility [13]. Therefore, the compound feed industry faces a dilemma: they must produce pellets with either a high PDI and a fine average particle size or with a low PDI using a coarse average particle size. Producing pellets with a combination of a satisfactory PDI and coarse particles present is therefore a challenge for the industry. 

Once the components of pellets are well ground, scaled and mixed, the pelleting set takes place. The objective of preconditioning is to provide moisture and heat, which depends on steam application and residence time, and this then promotes chemical changes that allow strong binding between feed particles when they pass through the pellet die holes. However, temperature and time must be controlled to avoid nutritional losses due to heat damage. Whereas water and steam are provided to the bulk via preconditioning [11,13,14], it is also possible to add shear and friction forces (mechanical stress) using the high-temperature–short-time (HTST) conditioner, the expander [14,15]. Expansion intensity, measured in kilowatt hours per ton (kWh/t), is adjusted by opening or closing the discharging gap, keeping the same throughput ratio. Expansion happens when the high-pressured mash achieves atmospheric conditions after being released; the pressurized steam inside the mash expands [15], contributing to achieving higher nutrient availability by disrupting fiber, denaturating protein and gelatinizing starch. The effects of HTST on digestibility and other changes were reviewed by Boroojeni et al. [16]. The authors described the contradictory effects on animal response when using HTST, such as inconsistency in starch, fat and amino acid (AA) digestibility, as well as the content of apparent metabolizable energy, corrected for nitrogen (AME_N_), for which it is hard to define the exact effects expected when this conditioning is used. It is unclear if the positive effects of expander conditioning prior to pelleting are observed mainly in the efficiency of the feed mill during the processing of compound feed and its pellet quality, or in broiler performance. In the case of higher broiler performance and carcass yields, these may be explained by alterations in particle size (secondary grinding), starch gelatinization, protein denaturation, organic matter digestibility and hygienization, which are associated with expander conditioning prior to pelleting [15]. 

Experiments evaluating the combination of different processing steps in compound feed production in relation to the evaluation of broiler performance have seldom been described. Thus, the objective of this study was to investigate the effects of a combination of mill types, particle sizes and expander conditioning prior to pelleting, applied to the same diet formulation of compound feed for broilers. Responses were obtained for the primary particle size of pellets and pellet quality to show the effects of the processing steps on the final product fed to broilers. In addition, the responses obtained by feeding these pellets represent the cumulative effects of the processing steps (mixing, grinding, preconditioning, pelleting and cooling), and were measured as broiler nutrient digestibility, gizzard development, performance and slaughter yield. We hypothesize that associating coarse particle size, ground using a roller mill and conditioned using an expander prior to pelleting, increases pellet quality as well as improving the growth and slaughter performance of broilers. 

## 2. Materials and Methods

### 2.1. Diets 

Two corn–SBM-based broiler diets were formulated using the analyzed amino acid content of corn and SBM (AMINONIR^®^ Advanced, Evonik Operations GmbH). The estimated digestibility of AA and AME_N_ content of corn and SBM for broilers was used for the calculation of the diet using digestibility coefficients based on AMINODat^®^ [17], as shown in Table 1. The nutritional content of these diets aimed to follow or exceed the recommendation of the GfE [18] for the grower- and finisher-broiler-rearing feed phases, respectively, using 13.37 and 13.25 MJ/kg of AME_N_; 22.64 and 21.34% of CP; 1.04 and 0.88% of standardized ileal digestible (SID) lysine. A common starter diet was used (12.35 MJ/kg of AME_N_, 22% of CP and 1.16% of SID lysine) from 1 to 7 days. The diet AA composition was also measured and is shown in Table 1.

### 2.2. Feed Processing Design

With the above diets, a 2 × 3 × 2 factorial feed processing experiment was designed, with the factors being 2 grinding mill types, 3 particle sizes and 2 mash conditionings prior to pelleting. Corn, SBM and grass meal were scaled following the feed formulations, and then, mixed in a ploughshare mixer (Lödige FKM 1200 D) for 2 min. These mixes were ground to obtain 0.8, 1.2 and 1.6 mm average particle sizes using a roller mill (LWM 400-1) or a horizontal hammer mill (Awila Typ AWF 20). Both mills were set to grind these mixes to the intended sizes. At the point when the intended particle sizes were achieved, they were then used as constituents of the compound feed, via mixing with the remaining ingredients, thus completing the diets. For the hammer mill, it was necessary to change the sieves, motor axis rotations and throughput to achieve the intended particle size. The adjustments for the 0.8 mm particle size were: 8 mm sieve, 50 Hz, and 1.76 t/h; for the 1.2 mm particle size: 15 mm sieve, 46 Hz, and 2 t/h; and for the 1.6 mm particle size: 15 mm sieve, 33 Hz, and 0.88 t/h of throughput. For the roller mill, it was necessary to change the gap between rollers, as well as the throughput, to achieve the intended particle size, keeping the same roller speed, i.e., 8 and 12 m/s. To obtain the 0.8 mm particle size, it was necessary to grind the mixture in a two-step mill setting. The first step was with a 1 mm gap and 0.85 t/h of throughput. The second one was with a 0.2 mm gap and 0.25 t/h of throughput. For the 1.2 mm and 1.6 mm particle sizes, only one mill setting was necessary: 0.3 mm and 0.56 t/h, and 1 mm and 0.85 t/h, respectively.

Afterwards, two processing lines were used to produce the diets. One of them used a steam pre-conditioning (85 °C and 14% moisture) step followed by expander (OE 15, Amandus Kahl, Germany) conditioning prior to pelleting, using 7 kWh/t SME (specific mechanical energy), and the expander heat temperature was 103 °C. The second processing line used only steam preconditioned (75 °C and 15.5% moisture) prior to pelleting. Pellets were produced using dies with holes with a 4 mm diameter and 24 mm length when the expander was used, and a 4 mm diameter and 40 mm length when only preconditioning was applied. Both die types were installed in a pellet press Type 33-390 (Amandus Kahl, Germany) working at 4 kWh/t, using a throughput of 1 t/h. The pellets were cooled on a belt cooler (Amandus Kahl, Germany), and then, crumbled in a LWM 100-1 using a 7.5 and 5 m/s roller speed and a 3 mm gap between them. Grower and finisher diets were used as replicates for the feed processing trial step, and both were produced once. 

### 2.3. Particle Size Distribution

To achieve the particle size distribution that was aimed for in the trial, non-ground corn, SBM and grass meal from each from the same analyzed batches were mixed prior to grinding following the calculated diets. Both machines, the hammer and roller mill, were adjusted to obtain average particles as close as possible to 0.8, 1.2 and 1.6 mm, as described above. The measure of dry mash particle size distribution in compound feed after mixing was performed by sifting the samples through a set of nine standardized Retsch^®^ sieves, nominally opened from top to bottom at 3.15, 2.50, 2.00, 1.40, 1.25, 1.00, 0.80, 0.50 and 0.20 mm, respectively. The sieve shaker was a Retsch^®^ (AS 200 Control^®^, Haan, Germany) using an adapted dry-sieving method [19]. In total, 10 subsamples were taken when the mixer was discharging to compose the sample of each mixed batch. To demonstrate secondary grinding, samples were taken from the compound diet in mash form and from the cooled pellets before crumbling.

The primary particle sizes of the pellets were measured using the wet-sieving method. The pellets were sampled immediately after cooling. A sieve tower was set with eight standardized Retsch^®^ sieves, nominally opened from top to bottom at 3.15, 2.00, 1.40, 1.00, 0.71, 0.50, 0.40 and 0.20 mm. 

The pellets (~100 g) were dissolved in one liter of water at room temperature for one hour. This solution was then discharged into the sieve tower and washed with 10 L of water. Afterwards, the sieves were dried at 105 °C for 4 h. The remaining samples in each sieve were weighed and used in the particle size calculator and expressed on a cumulative mass basis. D_10_, D_50_ and D_90_ are the Q3 distribution (range), and represent the proportions of 10%, 50% and 90% of the sample that were smaller than the size able to pass the sieve hole as indicated [20]. D_50_ is the average particle size. 

### 2.4. Pellet Quality

The PDI was measured using the *p*-fost method [21] and pellet hardness using an Amandus Kahl automatic pellet hardness tester, measuring the force (kg/cm²) necessary to promote the first fracture of each individual pellet. This measurement used the average hardness of 10 pellets per replication to provide a mean value for replication for use in further statistical analysis.

### 2.5. Broiler Experimental Design and Housing

A total of 864 unsexed one-day-old, Ross 308 broiler chickens, vaccinated against Marek’s disease, were placed in 36 pens (4 birds/m^2^, 12 per pen); each pen was littered with wood shavings, and equipped with one bell drinker, one trough feeder and one infrared lamp. The birds had *ad libitum* access to water and feed. The environmental temperature was controlled automatically using infrared lamps as heaters and extractor fans to ensure minimal air renewing and cooling. The temperatures used in the feeding trial followed the recommendations of the breeder: 32 °C at the chicken’s placement, and then, reducing this by 1 °C every two days until 21 °C was reached.

#### Performance Data

Body weight, FI and FCR were recorded at the end of each feed phase, by weighing the non-fasted animals and the feed that remained. Body weight gain and FCR were calculated for each phase and overall period. FCR was corrected using the weight of dead birds.

### 2.6. Slaughter, Organ Development and Commercial Cuts

On day 39, all the remaining broilers were fasted for 8 h, individually weighed, stunned by a percussive blow to the head, and then, bled through a jugular vein cut, scalded at 60 °C for 45 s and defeathered. Evisceration was manually performed, and the carcasses were statically chilled in a cooling room at 4 °C for 24 h. Livers and gizzards were taken and weighed. Commercial cuts were performed by a crew of industry-trained personnel, who produced bone-in legs, wings, as well as deboned breast fillets with tenders. Abdominal fat was weighed separately. Carcass yields and gizzards were expressed relative to live weight, while commercial cuts and abdominal fat were expressed relative to eviscerated carcass weight.

### 2.7. Determination of Apparent Total-Tract Digestibility (ATTD) and Metabolized Energy

Titanium dioxide (TiO_2_) was used in the finisher feed as a non-digestible marker. Partial excreta samples were taken, placing all the birds of each pen in metal cages with steel plates for two hours at 35 and 36 days of age. The excreta were stored in vacuum-sealed plastic bags at −20 °C, and then, freeze-dried for further analyses.

### 2.8. Determination of the Coefficients of Apparent Ileal Digestibility

At 38 days of age, 4 birds from each pen were randomly taken, stunned by a percussive blow to the head, then, bled through a jugular vein cut. Digesta samples were taken from the ileum, between the Merkel diverticulum and 3 cm cranial to the ileo-cecal junction by flushing with distilled water. These were stored in vacuum-sealed plastic bags at −20 °C, and then, freeze-dried for further analyses.

### 2.9. Chemical Analysis

The finisher diet was analyzed for dry matter (DM) (method 3.1.4), ether extract after acid hydrolysis (method 5.1.2) and starch (method 7.2.1) according to the standard procedures of VDLUFA [22]. Nitrogen was determined via the Dumas method (method 968.06 [23]) (Büchi, DuMaster D-480, Flawil, Switzerland). The gross energy content was determined by a calorimeter calibrated with benzoic acid as a standard (IKA C 200, Parr instruments, Staufen, Germany), and the amino acid content was determined via ion-exchange chromatography with post-column derivatization with ninhydrin, as described by Figueiredo-Silva et al. [24]. Ca and P were analyzed using an atomic absorption spectrometer (PerkinElmer, Analyst 200). TiO_2_ was analyzed following Jagger et al. [25]. In the grower diet, only DM, AA, Ca and P were analyzed.

### 2.10. Calculations

Total-tract digestibility and AME_N_ were calculated using the equations suggested by Kong & Adeola [26]:Digestibility (%) = [1 − (M_d_/M_e_) * (E_d_/E_e_)] * 100,(1)
AME_N_ (MJ/kg) = GE_d_ − [GE_e_ * (M_d_/M_e_)] − 0.0344 * {N_d_ − [N_e_ * (M_d_/M_e_)]},(2)
where M_d_ represents the concentration of TiO_2_ in the diet in g/kg; M_e_ represents the concentration of TiO_2_ in the excreta and ileal digesta in g/kg; and E_d_ represents the content of total fat, starch and amino acids in g/kg and gross energy (MJ/kg) in the diet. E_e_ represents the amount of total fat (g/kg) in the excreta, and starch and amino acids (g/kg) in the ileal digesta. GE_d_ represents the gross energy of the diet (MJ/kg), GE_e_ represents the gross energy of the excreta (MJ/kg). N_d_ represents the nitrogen (g/kg) in the diet and N_e_ represents the nitrogen (g/kg) in the excreta. The values of all the nutrients and energy were expressed on a DM basis for the calculations.

### 2.11. Statistical Analysis

The pen was used as the experimental unit. The experiment design was a completely randomized factorial arrangement with 2 mills (hammer and roller), 3 particle sizes (0.8, 1.2 and 1.6 mm) and 2 expander conditionings prior to pelleting (with and without). All the data were subjected to normality testing using the Shapiro–Wilk test [27] prior to the 3-way ANOVA using the GLM procedures from SAS Institute [28]. When significant, the LSmeans were separated using the Tukey–Kramer test [29] and accepted as different when *p <* 0.05.

The model used was:Y_ijkl_ = µ + γ_i_ + α_j_ + β_k_ + δ_l_ + (αβ)_jk_ + (αδ)_jl_ + (βδ)_kl_ + (αβδ)_jkl_ + ε_ijkl_(3)
where: Y_ijkl_ = observation, µ = population mean, γ = period effect (_i_ = 1, 2), α_j_ = mill effect (_j_ = hammer, roller), β_k_ = particle size effect (_k_ = 0.8, 1.2, 1.6), δ_l_ = expander effect (_l_ = with, without), (αβ)_jk_ = interaction between mill and particle size effect, (αδ)_jl_ = interaction between mill and expander effect, (βδ)_kl_ = interaction between particle size and expander effect, (αβδ)_jkl_ = interaction between mill, particle size and expander effect and ε_ijkl_ = residual error.

The mash diet particle size (dry sieving) statistical analysis followed the model below:Y_ijk_ = µ + γ_i_ + α_j_ + β_k_ + (αβ)_jk_ + ε_ijk_(4)
where: Y_ijk_ = observation, µ = population mean, γ_i_ = block effect (_i_ = grower, finisher), α_j_ = mill effect (_j_ = hammer, roller), β_k_ = particle size effect (_k_ = 0.8, 1.2, 1.6), (αβ)_jk_ = interaction between mill and particle size effect and ε_ijk_ = residual error. The above model was also used when just two factors (mill and particle size) were present. The compound feed for growers and finishers were used as blocks (r = 2).

The statistical analysis for pelleted diet particle size (wet sieving) followed the model below:Y_ijkl_ = µ + γ_i_ + α_j_ + β_k_ + δ_l_ + (αβ)_jk_ + (αδ)_jl_ + (βδ)_kl_ + (αβδ)_jkl_ + ε_ijkl_(5)
where: Y_ijkl_ = observation, µ = population mean, γ_i_ = block effect (i = grower, finisher), α_j_ = mill effect (j = hammer, roller), β_k_ = particle size effect (k = 0.8, 1.2, 1.6), δ_l_ = expander effect (l = with, without), (αβ)_jk_ = interaction between mill and particle size effect, (αδ)_jl_ = interaction between mill and expander effect, (βδ)_kl_ = interaction between particle size and expander effect, (αβδ)_jkl_ = interaction between mill, particle size and expander effect and ε_ijkl_ = residual error. The compound feed for growers and finishers were used as blocks (r = 2).

## 3. Results

### 3.1. Particle Size Distribution, Power Applied in the Feed Mill and Pellet Quality

No interactions were observed for particle size distribution when mill type, particle size (dry sieving) and expander conditioning prior to pelleting (only wet sieving) were analyzed in the mash diets (Table 2). The intended particle sizes (Q3 = 50%, which represents the average particle size) for the trial were achieved for 0.8 mm and almost for 1.2 mm and, on average, were 0.4 mm finer for the 1.6 mm group, as shown in Table 2 and Figure 1. The biggest difference for mill type was observed in D_90_ of the Q3 distribution (dry sieved). Closer examination of the coarse particles in the mash diets (Q3 = 90%) showed that the roller mill produced more uniform particles than the hammer mill (*p* = 0.038). Roller mill D_90_ particle distribution was closer (1.971 → 1.094 mm) to the D_50_ particles than hammer mill D_90_ particles (2.589 → 1.000 mm). In addition, the 1.6 mm group had higher particles sizes at D_90_ than 1.2 mm and 0.8 mm, at 2.829, 2.270, and 1.742 mm, respectively. The particle size distribution of the mash was measured from the compound feed sampled during mixer discharge, showing that the 1.6 mm particle size group was more highly influenced by the fine ingredients (e.g., premixes).

As expected, wet-sieved pellets revealed that particle sizes at Q3 = 50% of the 0.8 mm group were finer than those of the 1.6 mm group (*p* = 0.031) after pelleting (Table 2). Further, expander conditioning and pelleting intensely contributed to secondary grinding regardless of the particle size or mill type.

The reduction in particle size is remarkable when compared to the intended particle size at grinding. The sizes of particles ingested by the birds were lower than originally intended, highlighting the secondary grinding effect of the processing line.

The use of the expander prior to pelleting clearly resulted in higher energy consumption (kWh/t) (Table 3). Pellet hardness, measured as the N/cm^2^ necessary to break them, was affected by the expander. The use of the expander produced pellets that needed less strength to break, but fewer fine particles (particles finer than 2.8 mm, sieved immediately after pellet cooling) were produced after using the expander (Table 3). The mill type and PS did not have any effect on pellet quality. Mill type and particle size did not change the proportion of fines. The PDI was affected by the interaction between the expander and particle sizes, the results for which are shown in Table 4.

### 3.2. Broiler Performance

The effects on broiler performance were observed and the results are shown in Table 5 (*p* < 0.05). In the finisher phase, feed intake (FI) was affected by expander conditioning. A similar reduction was observed for FI in the overall trial. Similarly, few interactions were observed for all the feed phases and for the overall growing period in the broiler performance results. The FI in the grower phase was affected by the interaction between mill type and expander conditioning prior to pelleting, where the combination of the hammer mill with the expander resulted in about a 3% lower FI (Table 6). The feed conversion ratio (FCR) was affected in a three way-interaction, as shown in Table 5 and Table 6, where the combination of the roller mill, 1.6 mm particles and expander conditioning prior to pelleting showed the most efficient FCR, which was 1.511 g:g for the overall growing period.

### 3.3. Broiler Slaughter Performance and Gizzard Development

Although carcass yield was affected by the three-way interaction (Table 7), the effect was less pronounced among treatments. About a 2% lower carcass yield was obtained only when the major ingredients were processed to obtain a 0.8 mm particle size, using the roller mill with the complete mash diet that was expanded prior to pelleting (Table 8). Gizzard relative weight was just affected by particle size as the main effect. The feed with a 1.6 mm particle size resulted in heavier gizzards than the ones of the birds fed 0.8 mm particles (Table 7), as expected. As a trend (*p* = 0.057), expander processing prior to pelleting decreased gizzard relative weight.

Commercial cut yields were not affected by the treatments (Table 7).

For abdominal fat proportions, a two-way interaction was observed for the treatments mill type and particle size, as well as expander conditioning prior to pelleting and particle size (Table 7). When expander conditioning prior to pelleting and particle size interacted, the abdominal fat proportion was greater for the 0.8 mm processed compound feed using expander conditioning prior to pelleting. This effect was not observed for the feeds processed to obtain particle sizes of 1.2 and 1.6 mm. Considerations of mill type and particle size indicate that the lowest amount of abdominal fat was obtained when the roller mill was associated with 1.2 mm particle size (Table 8).

### 3.4. Broiler Nutrient Digestibility and Energy Utilization

The apparent ileal digestibility (AID) of starch and amino acids were positively affected by using expander conditioning prior to pelleting (Table 9). Mill type and particle size did not show effects on the AID of starch and amino acids. No interactions were observed.

The apparent metabolizable energy (AME_N_) content was affected by mean particle size (Table 10). The AME_N_ content was also affected by the interaction between mill type and expander conditioning prior to pelleting (Table 10 and Table 11). The interaction showed that the combination of roller mill and pelleting without expander conditioning prior to pelleting resulted in the lowest AME_N_ content.

The OM digestibility was affected by particle size, with the highest broiler digestibility for the 1.2 mm mean particle size (Table 10). Furthermore, the digestibility of OM was affected by the interaction between mill type and expander prior to pelleting, where the lowest digestibility (71%) was observed in the broilers that received the diets processed in the line with the roller mill combined with preconditioning only, without expander conditioning prior to pelleting (Table 11). The crude fat digestibility was affected by the interaction between expander conditionings prior to pelleting and particle size. For the particle sizes of 0.8 and 1.2 mm, the use of the expander prior to pelleting improved fat digestibility (Table 11).

## 4. Discussion

### 4.1. Experimental Setup

A challenge in experiments such as that described here, involving both feed processing treatments and animal performance, is establishing whether there is a strong causal relationship between the effects generated during the different processing steps and the animals’ responses. 

In the present study, the three different factors of feed processing, i.e., grinding type, particle size and expander conditioning prior to pelleting, were cumulative, being expressed in the pellets in the final step in processing (Figure A1). However, only pellets were fed to the birds. In such a situation, animal performance and gizzard development are the result of cumulative effects of each individual processing step. These cumulative compound feed processing steps may mask carry-over effects among the different processing steps, as the animal consumes only the final product of the processing line, i.e., the pellet (with crumbling to adjust the pellet size to the beak size of the birds). In addition, fixing the effects of processing without considering the animal response may neglect those carry-over effects; for example, grinding equipment provides size reduction, but the final compound feed fed to birds is also a result of the different processing effects such as secondary grinding or starch gelatinization. Furthermore, a lack of animal response should not be a reason to avoid the investigation of such processing steps; much of the efficiency in processing compound feed may not be transferred to animal performance, but only in process efficiency at the feed plant, such as improvements in pellet quality. Finally, it should be kept in mind that the design of the processing line may have contributed to the observed effects, and the results cannot necessarily be applied directly or/and proportionally to processing lines with higher production capacity (in t/h) due to the individual features of each machine within a processing line.

### 4.2. Processing Effects

Few studies have evaluated a combination of different factors in the processing line in terms of effects on feed quality and animal response [30]. Without significant interactions, analyses of each factor as a main effect are important in terms of understanding the major impact that this one factor independently has on animal response.

In our study, taking the mean particle size after grinding or after pelleting, few effects were observed when mill types (main effects) were compared. Roller mills produced higher uniform particles than hammer mills, as reported previously by Koch [7]. This explains the Q3 > 90% particles being narrower than the average particles (Table 2). Svihus et al. [2] reported that the amounts of extremely fine and coarse particles were reduced when a roller mill was used, compared with using a hammer mill for the same intended particle size. Including the further processing steps, preconditioning and pelleting of the mash diets, the mill type effect on particle size disappeared. In addition, a more intense reduction in coarse particles during the expander conditioning and pelleting processes occurred (Table 2), supporting the occurrence of secondary grinding through the processing of the diets [20]. This processing effect seems to be substantial, and therefore, cannot be neglected further, especially in the context of the interpretation of causal relationships between particle size gradients obtained with grinding and the animal response. On the other hand, the results show that intense grinding is not necessary when secondary grinding is expected. Grinding to large particle sizes may enable reduced energy consumption in the processing line. However, depending on the equipment used in the processing line, the contribution of secondary grinding is not easy to estimate.

### 4.3. Animals’ Responses to Particle Size Reduction

The digestibility of amino acids and starch is an important aspect to consider with regard to the evaluation of animals’ responses to changes in the processing line in relation to the nutritive value of compound feed. Our study showed few effects (Table 9). This lack of effect is probably related to the secondary grinding effect in the processing line, which almost removed the differences intended by grinding (i.e., 0.8, 1.2 and 1.6 mm), in the crumbled pellets offered to the birds (Figure 1). This means that the differences between the observed particle sizes in the pellets were similar, resulting in similar digestibility.

Only the use of the expander prior to pelleting increased the amino acid digestibility, and only the digestibility of EEh was influenced by the interaction between expander conditioning prior to pelleting and particle size (Table 11). The rupture of the cell matrix resulting from using the expander probably contributed further to increasing EEh and OM digestibility. However, at the higher intended particle size of the major ingredients of the compound feed (1.6 mm), the use of the expander was not able to increase EEh digestibility (Table 11). Amerah et al. [31] used diets based either on corn or wheat for broilers and reported a strong tendency toward an interaction (*p* < 0.06) of grain type and particle size for metabolized energy. The coarse grinding of corn resulted in lower AME_N_ content using corn-based diets in their study, findings that also agree with those from our study.

Using large particles in compound feed is just one aspect of expander conditioning prior to pelleting when the aim is to improve animal response. In the present study, the effect on particle size reduction was very intense (see Table 2), showing that the friction and shear forces applied to the mash in the expander barrel, and subsequently passing to the pellet press, changed the particle size (Figure 1). This effect is well known and is well documented in the literature, as shown by Fancher et al. [15], Campbell et al. [32], Liermann et al. [33] and Lyu et al. [20]. The secondary grinding coming from the expander and the pellet press independently, or the two effects in combination, unfortunately, cannot be separated in the present study. However, the results show that expander conditioning prior to pelleting intensifies secondary grinding, at least for the processing line used in the present study (Table 2). In addition to secondary grinding, using expander conditioning prior to pelleting in the processing line implies higher SME consumption (expressed in kWh/t), but the expander reduced energy consumption by the pellet press and improved pellet quality (Table 3), as also observed by Fancher et al. [15]). Although higher energy consumption (see Table 3 and Table 4) was observed, it is expected that the improved animal response from using expander conditioning prior to pelleting must compensate (at least economically) for this higher electricity consumption. The compensation occurred partially; however, it should be made clear that the effects of expander conditioning prior to pelleting on animal response are not solely responsible for the decision to use this HTST technology, as the positive effects on savings in the processing line per se must be considered (for instance, less wear and tear of the pelleting die, a higher output rate (t/h) of the pellet press and the inactivation of anti-nutritive factors and hygienization effects). In the present study, the results showed that conditioning using the expander prior to pelleting positively affected amino acid and starch ileal digestibility, which were 0.01 to 0.02% points higher (Table 9), and the broiler digestibility of OM and EEh (Table 11). The processing line should be arranged with the consideration that secondary grinding may occur, also affecting digestibility.

Although slightly reduced FI was observed for the overall period when expander conditioning prior to pelleting was used, the FCR (three-way interaction) was more effective when the expander was used (Table 5 and Table 6, Figure 1). The improved FCR is in accordance with previous findings [34], where expander conditioning improved the broiler nutrient digestibility of corn-based diets. The higher surface area promoted by the expansion of the particles is probably the reason for this improvement, improving conditions for enzymes to act in the digesta.

Performance at slaughter was less affected by the processing of compound feed. Some effects, such as the higher proportion of abdominal fat or gizzard weight, were expected and observed. The higher abdominal fat may have resulted from the higher uniformity and surface area of the particle size produced by the roller mill, which required less activity from the gizzard, resulting in more energy available to be converted into abdominal fat. In addition, the additional abdominal fat may be explained by the higher AME_N_ available after expander conditioning prior to pelleting, as measured in this trial. This higher AME_N_ should be balanced with the amino acid in the diet. As the expanded diets were not calculated to have higher amino acid concentration, the induced unbalanced ratio of lysine to AME_N_ may help to accumulate more abdominal fat, as also observed in previous studies [34].

The lack of effects on commercial cuts demonstrates that a still-balanced supply of amino acids in the diet, and no severe deficiency or damage due to processing conditions, were apparent when commercial cuts were used as the reference. As a consequence, it is necessary to adjust the Lys: AME_N_ ratio in diet formulation when the expander is used.

The gizzard’s relative weight is a good parameter to support the presence of larger particles in compound feed [11,16,31,35]. Although varying from 0.8 to 1.6 mm in particle size, designed to be obtained after the grinding step in the processing line, neither the mill type nor use of the expander prior to pelleting affected the gizzard size; this supports secondary grinding as an effect during processing in reducing large particles.

### 4.4. Secondary Grinding

Measuring particle size after the milling procedure did not include the secondary grinding effect, which occurs when the entire process, i.e., from mixing to pelleting, is considered. It is astonishing that processed compound feed for poultry, in which their major ingredient was ground into particle sizes of 0.8, 1.2 or 1.6 mm, ended up in pellets with average particle sizes ranging from 0.2 to 0.8 mm, a reduction of about 50 to 75% (Figure 1). This finding suggests that particle size should be measured using the pellets, as the role of particle size is well known in gizzard development [1].

The explanation of animal response does not include the possible effect of secondary grinding, as the designed particle size was used as an independent variable. As already mentioned, the intended particle sizes were not the ones that the birds effectively ate. The size reduction due to secondary grinding was more intense in the compound feed with coarse particles. Closer examination of each category showed that the 0.8 mm particles were reduced 2.0 times, the 1.2 mm were reduced 2.6 times and the 1.6 mm were reduced 2.3 times (Table 2). Similar results were also observed by Abdollahi et al. [11]. It is known that secondary grinding occurs, but previous studies did not measure its magnitude on particle size reduction [11,16]. As the pellet quality did not change due to the particle size effect as the main effect (Table 3), this supports the effect of secondary grinding; it shows that larger particles may be used in compound feed production, as they are reduced in size during processing by secondary grinding, without compromising the integrity of the pellets. However, as shown in Table 4, the SME necessary to process compound feed with larger particles (1.6 mm) required lower kWh/t (18.0 kWh/t) when the expander was used prior to pelleting, compared to the particle sizes 0.8 and 1.2 mm. Without expanding prior to pelleting, more SME (7.9 kWh/t) was necessary to process the feed with a 1.6 mm particle size compared to that with 1.2 and 0.8 mm particle sizes. These findings suggest that grinding to obtain a coarser particle size using a roller mill and expanding prior to pelleting are interesting because lower kWh/t is necessary to process it. In contrast, larger particle sizes remaining in pellets (Q3 = 50%, 0.654 mm, Table 2) were observed when preconditioning and pelleting were used. Large particles enhance the functionality of broilers’ gastrointestinal tracts, as reported in the literature [14,16,31,36], and this also agrees with the relative gizzard size in our study (Table 7).

Despite this huge variation, processing compound feed with the aim of a 1.6 mm mean particle size may have to be seen as a future strategy for poultry nutrition in corn/soybean-based diets. Several results in the present study support this strategy. Whereas the effect of mill type did not influence AID or ATTD, as also observed by Nir et al. [35], the effect of particle size demonstrated substantial effects on AME_N_ content, as well as the ATTD of organic matter and crude fat. However, using expander conditioning prior to pelleting is necessary to improve PDI, especially when the aim is grinding for coarse particle size, despite the contribution of secondary grinding.

## 5. Conclusions

The study aimed to demonstrate causal relationships of compound feed processing steps with animals’ responses. However, the study revealed a cumulative effect of the different processing steps (milling, mixing, conditioning and pelleting). The cumulative effect influenced the pellets obtained from the processing line differently, and the most significant effect was secondary grinding. However, it should be kept in mind that the design of the processing line may have contributed to this, and the results cannot necessarily be applied directly to larger-scale processing with a higher production capacity (in t/h). It is possible that the contact of each kernel or particle with the mechanics through upscaling of the processing line may even reduce the prediction of secondary grinding.

In addition, it is important to mention that variations in the processing line due to mill type, particle size and conditioning with the expander prior to pelleting may improve pellet quality or result in efficient use of the applied kWh/t, but may not necessarily influence animal response. However, the influence on pellet quality and efficient resource utilization within the compound feed plant should be considered and prioritized. On the other hand, this trial positively shows that there were broiler responses, such as FCR and AME_N_, suggesting that when diets were conditioned using the expander prior to pelleting, the calculated energy content of the diets (or the ingredients individually) may be adjusted due to the higher availability of AME_N_ after expander conditioning. In general, the processing line using the roller mill, coarse particle size and expander conditioning prior to pelleting contributed to high pellet quality, improved nutrient digestibility and feed efficiency of the broilers. This combination in the processing line can therefore be recommended for feed mill practice.

## Figures and Tables

**Figure 1 animals-12-02707-f001:**
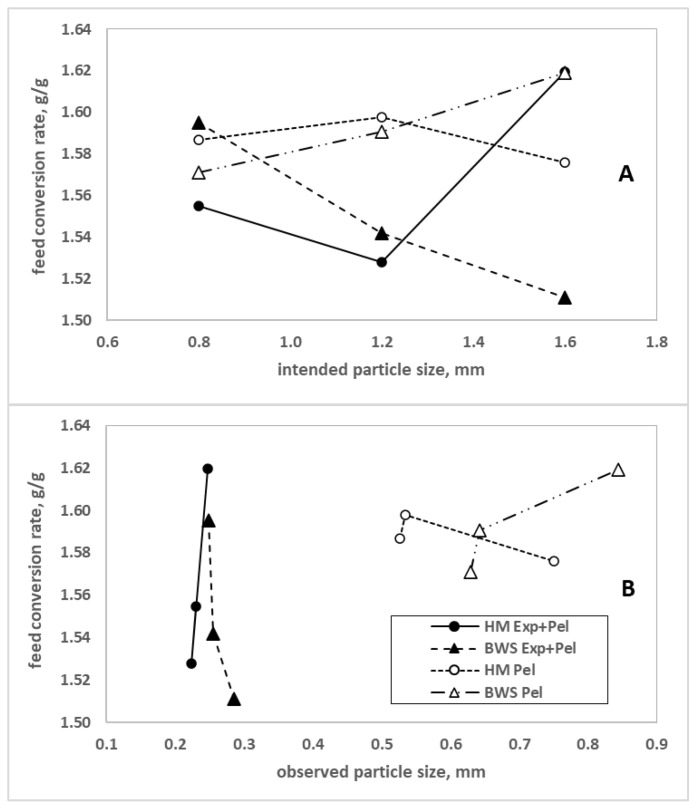
Comparison between intended and observed particle sizes. (**A**) Intended particle sizes are 0.8, 1.2 and 1.6 mm. (**B**) The particle sizes observed after sampling in the feeder, using wet-sieve analyses, representing the compound feed offered to the birds. Each line corresponds to all of the processing steps. HM: hammer mill; RM: roller mill; Exp: expander; Pel: pelleting.

**Table 1 animals-12-02707-t001:** Ingredients and nutrient composition of experimental broiler diets on an as-is basis.

Item	Grower (8 to 21 d)		Finisher (22 to 38 d)	
Corn	519.0		571.0	
SBM HP, 48% CP	364		288.3	
Soybean oil	67.4		51.3	
Grass meal	20.0		30.0	
Corn gluten			30.0	
Dicalcium phosphate	13.4		11.4	
Calcium carbonate	9.7		10.0	
NaCl	2.7		2.5	
Vitamin + Mineral premix ^1^	1.0		1.0	
DL-Methionine	1.2		1.0	
L-Lysine HCl	0.2			
Choline chloride	0.8		0.4	
Coccidiostat ^2^	0.5			
Phytase ^3^	0.1		0.1	
Titanium dioxide (TiO_2_)			3.0	
Analyzed or calculated (*) nutrient composition, in g/kg unless specified differently				
Dry matter	921.6		920.4	
AME_N_, MJ/kg	13.37 *		13.25 *	
Crude protein	233.3		225.1	
Total fat (EEh)	100.4		70,9	
Calcium	8.9		8.6	
Available phosphorus	4.6 *		4.1 *	
Sodium	1.2 *		1.2 *	
Chloride	2.3 *		2.2 *	
Potassium	9.5 *		8.4 *	
Choline, mg/kg	1733 *		1362 *	
Amino acid composition	Total	Digestible	Total	Digestible
Met	4.4	4.2 *	4.9	4.1 *
Lys	12.3	10.4 *	11.6	8.8 *
Met + Cys	8.0	7.0 *	8.3	6.1 *
Thr	8.6	7.0 *	8.3	6.6 *
Trp	2.7 *	2.4 *	2.4 *	2.0 *
Arg	15.5	13.7 *	14.9	12.1 *
Ile	10.2	8.6 *	9.8	8.0 *
Leu	19.3	16.9 *	18.8	17.8 *
Val	11.0	9.2 *	10.6	8.8 *

^1^ Supplied per kilogram of diet: vitamin A, 10,000 IU; vitamin D_3_, 4000 IU; vitamin E, 20 IU; vitamin K_3_, 4 mg; thiamine, 3 mg; riboflavin, 7.5 mg; pyridoxine, 4.5 mg; cyanocobalamin, 0.0225 mg; pantothenic acid, 19.5 mg; niacin, 69 mg; folic acid, 0.195 mg; biotin, 0.012 mg; iron, 16.8 mg; zinc, 80 mg; manganese, 100 mg; copper, 12 mg; iodine, 1 mg; selenium, 0.25 mg. ^2^ Sodium monensin, 200 g/kg, Elanco Animal Health, Greenfield, NY, USA. ^3^ Optiphos^®^ 2500 FTY, Huvepharma EOOD, Sofia, Bulgaria. * Calculated values.

**Table 2 animals-12-02707-t002:** Estimated particle size cumulative distribution (mm) of broiler diets ground using two types of mill at three particle sizes (PS), submitted to conditioning with or without expander (E + P) prior to pelleting, and the quality of resulting pellets.

Item	Mash (Dry Sieving)			After Pelleting (Wet Sieving)		
	Q3 Distribution ^1^, mm					
	10%	50%	90%	10%	50%	90%
Mill type						
Hammer	0.370	1.000	2.589 ^a^	0.050	0.418	1.894
Roller	0.346	1.094	1.971 ^b^	0.051	0.483	1.691
PS, mm						
0.8	0.313 ^b^	0.820 ^c^	1.742 ^b^	0.053	0.408 ^b^	1.512 ^b^
1.2	0.366 ^ab^	1.081 ^b^	2.270 ^ab^	0.050	0.413 ^ab^	1.641 ^b^
1.6	0.395 ^a^	1.241 ^a^	2.829 ^a^	0.050	0.531 ^a^	2.226 ^a^
E + P						
With				0.043 ^b^	0.248 ^b^	1.430 ^b^
Without				0.058 ^a^	0.654 ^a^	2.156 ^a^
SEM	0.031	0.062	0.404	0.008	0.090	0.285
*p*-value						
Mill	0.230	0.054	0.038	0.640	0.100	0.110
PS	0.025	<0.001	0.025	0.700	0.031	<0.001
E + P				<0.001	<0.001	<0.001
Mill*PS	0.530	0.460	0.840	0.780	0.990	0.190
Mill*E + P				0.540	0.350	0.330
PS*E + P				0.790	0.095	0.300
Mill*PS*E + P				0.690	0.990	0.420

^a > b > c^ LSmeans in column with different superscripts differ significantly according to Tukey–Kramer test, *p <* 0.05; ^1^ the Q3 distribution represents the proportions of 10%, 50% and 90% of the sample that are smaller than the size needed to pass the sieve hole as indicated.

**Table 3 animals-12-02707-t003:** Power applied to broiler diets ground using two mill types at three particle sizes (PS), submitted to conditioning with or without expander (E + P) prior to pelleting, and the quality of the resulting pellets.

Item	kWh/t ^1^	Pellet Quality		
	Exp + Pel ^2^	PDI ^3^, %	Hardness, N/cm^2^	Fines ^4^, %
Mill type				
Hammer	13.4	83.9	16.4	6.4
Roller	12.5	83.3	16.8	6.5
PS, mm				
0.8 ^5^	13.0	84.6	15.8	6.5
1.2 ^6^	12.9	85.0	16.7	6.4
1.6 ^6^	12.9	81.1	17.2	6.5
E + P				
With	18.7	86.0	14.4 ^b^	5.1 ^b^
Without	7.25	81.1	18.8 ^a^	7.9 ^a^
SEM	0.46	3.10	2.38	0.63
*p*-values				
Mill	<0.001	0.670	0.680	0.600
PS	0.920	0.055	0.520	0.970
E + P	<0.001	0.002	<0.001	<0.001
Mill*PS	0.560	0.380	0.520	0.380
Mill*E + P	0.009	0.380	0.510	0.076
E + P*PS	0.001	0.006	0.540	0.860
Mill*PS*E + P	0.130	0.470	0.620	0.710

^a > b^ LSmeans in column superscripted with different letters differ significantly according to Tukey–Kramer test, *p <* 0.05; ^1^ kWh/t = kilowatt hour per ton for the specific mechanical energy (SME) input; ^2^ Exp + Pel = sum of specific mechanical energy input of expander conditioner and pellet press; ^3^ PDI = pellet durability index; ^4^ Fines = percentage of fine particles in pellets sieved using a 2.8 mm sieve; ^5^ hammer mill with 8 mm sieve, roller mill with two pairs of rollers (see Figure A1); ^6^ hammer mill with 15 mm sieve, roller mill with a single pair of rollers (See Figure A1).

**Table 4 animals-12-02707-t004:** Breakdown of significant interactions as presented in Table 3.

Item		kWh/t	Pellet Quality
		E + P	PDI, %
Mill type	E + P		
Hammer	With	18.9 ^dD^	
	Without	8.0 ^dE^	
Roller	With	18.5 ^dD^	
	Without	6.5 ^eE^	
E + P	PS, mm		
With	0.8	19.4 ^Vv^	84.6 ^Vv^
	1.2	18.6 ^Vv^	86.3 ^Vv^
	1.6	18.0 ^Vw^	87.2 ^Vv^
Without	0.8	6.6 ^Ww^	84.6 ^Vv^
	1.2	7.3 ^Wvw^	83.7 ^Vv^
	1.6	7.9 ^Wv^	75.0 ^Ww^

^d > e^ Comparison between mill types within E + P, *p <* 0.05; ^D > E^ comparison between E + Ps within mill type, *p <* 0.05; ^V > W^ comparison between E + Ps within PS, *p <* 0.05; ^v > w^ comparison between PSs within E + P, *p <* 0.05.

**Table 5 animals-12-02707-t005:** Performance of broilers fed diets ground with two types of mill at three particle sizes (PSs), submitted to conditioning with or without expander (E + P) prior to pelleting, measured from 8 to 38 days (g).

Item	Grower, 8–22 d			Finisher, 23–38 d			Overall, 8–38 d		
	FI	BWG	FCR	FI	BWG	FCR	FI	BWG	FCR
Mill type									
Hammer	1093	807.3	1.350	2745	1622.2	1.690	3841	2429.5	1.580
Roller	1117	814.8	1.355	2795	1661.0	1.666	3915	2475.8	1.571
PS, mm									
0.8	1106	810.9	1.347	2786	1648.5	1.690	3897	2459.3	1.577
1.2	1094	807.0	1.349	2749	1644.0	1.671	3844	2451.0	1.567
1.6	1115	815.3	1.360	2775	1632.3	1.674	3894	2447.7	1.581
E + P									
With	1100	808.9	1.345	2736 ^b^	1636.7	1.665	3839 ^b^	2445.7	1.558
Without	1110	813.2	1.359	2804 ^a^	1646.4	1.691	3917 ^a^	2459.7	1.590
SEM	39.2	34.07	0.051	131.9	94.10	0.08	165.5	108.9	0.06
*p*-value									
Mill	0.013	0.350	0.600	0.110	0.085	0.230	0.063	0.076	0.720
PS	0.190	0.700	0.630	0.620	0.830	0.700	0.460	0.930	0.660
E + P	0.230	0.600	0.260	0.033	0.660	0.190	0.049	0.590	0.045
Mill*PS	0.100	0.390	0.920	0.520	0.240	0.150	0.380	0.600	0.460
Mill*E + P	0.034	0.800	0.290	0.520	0.710	0.620	0.380	0.350	0.430
E + P*PS	0.710	0.460	0.810	0.460	0.810	0.120	0.600	0.860	0.360
Mill*PS*E + P	0.240	0.770	0.640	0.090	0.410	0.080	0.087	0.530	0.019

^a > b^ LSmeans in column superscripted with different letters differ significantly according to Tukey–Kramer test, *p <* 0.05.

**Table 6 animals-12-02707-t006:** Breakdown of significant interactions as presented in Table 5.

Item		FI, 8–22 d				FCR, 8–38 d
Mill type	E + P		Mill	PS, mm	E + P	
Hammer	With	1078 ^eE^	Hammer	0.8	With	1.553 ^XaA^
	Without	1108 ^dD^		1.2		1.528 ^XaA^
Roller	With	1122 ^dD^		1.6		1.621 ^XaA^
	Without	1112 ^dD^		0.8	Without	1.588 ^XaA^
				1.2		1.600 ^XaA^
				1.6		1.575 ^XaA^
			Roller	0.8	With	1.596 ^XaA^
				1.2		1.543 ^XaA^
				1.6		1.511 ^XbB^
				0.8	Without	1.570 ^XaA^
				1.2		1.590 ^XaA^
				1.6		1.620 ^XaA^

^d > e^ Comparison between mills within E + P, *p <* 0.05; ^D > E^ comparison between E + Ps within mill, *p <* 0.05. ^X > Y^ comparison between mills within PS and E + P, *p <* 0.05; ^a > b^ comparison between PSs within E + P and mill, *p <* 0.05; ^A > B^ comparison between E + Ps within mill and PS, *p <* 0.05.

**Table 7 animals-12-02707-t007:** Slaughter performance and gizzard weight of broilers fed diets ground using two types of mill at three particle sizes (PS), submitted to conditioning with or without expander (E + P) prior to pelleting, measured at 39 days of age (%).

Item	Carcass ^1^	Gizzard ^1^	Abd. Fat ^2^ *	Breast ^2^	Legs ^2^	Wings ^2^
Mill type						
Hammer	80.7	1.06	1.58	29.1	25.2	9.38
Roller	81.0	1.05	1.53	29.5	24.8	9.26
PS, mm						
0.8	80.9	1.02 ^b^	1.56	29.2	25.4	9.28
1.2	81.0	1.06 ^ab^	1.54	29.5	24.8	9.26
1.6	80.8	1.10 ^a^	1.57	29.2	24.8	9.41
E + P						
With	80.7	1.04	1.61	29.1	25.0	9.32
Without	81.0	1.07	1.50	29.5	25.0	9.32
SEM	0.64	0.08	0.13	1.72	1.39	0.63
*p*-value						
Mill	0.039	0.450	0.082	0.380	0.230	0.480
PS	0.410	0.020	0.850	0.760	0.280	0.680
E + P	0.045	0.057	0.001	0.390	0.720	0.990
Mill*PS	0.016	0.230	0.012	0.520	0.960	0.150
Mill*E + P	0.660	0.260	0.870	0.440	0.270	0.130
E + P*PS	0.007	0.220	0.048	0.480	0.800	0.990
Mill*PS*E + P	0.028	0.130	0.310	0.850	0.740	0.940

^a > b^ LSmeans in column superscripted with different letters differ significantly according to Tukey–Kramer test, *p <* 0.05; ^1^ carcass and gizzard weights relative to live body weight; ^2^ Abd. fat, breast, legs and wings weights as percentage of cold carcass weight; * Abd. fat = abdominal fat.

**Table 8 animals-12-02707-t008:** Breakdown of significant interactions as presented in Table 7.

Item			Carcass, %			Abd. Fat, %
Mill type	PS, mm	E + P		Mill	PS, mm	
Hammer	0.8	With	80.7 ^aAm^	Hammer	0.8	1.57 ^yY^
	1.2		80.3 ^aAm^		1.2	1.64 ^yY^
	1.6		80.8 ^aAm^		1.6	1.54 ^yY^
	0.8	Without	81.3 ^aAm^	Roller	0.8	1.55 ^yY^
	1.2		81.1 ^aAm^		1.2	1.45 ^yZ^
	1.6		80.1 ^aAm^		1.6	1.59 ^yY^
				E + P	PS	
Roller	0.8	With	80.0 ^aAn^	With	0.8	1.66 ^vV^
	1.2		81.5 ^aAm^		1.2	1.55 ^vV^
	1.6		81.0 ^aAm^		1.6	1.63 ^vV^
	0.8	Without	81.3 ^aAm^	Without	0.8	1.45 ^vW^
	1.2		81.2 ^aAm^		1.2	1.54 ^vV^
	1.6		81.1 ^aAm^		1.6	1.50 ^vV^

^a > b^ Comparison between mills within PS and E + P, *p <* 0.05; ^A > B^ comparison between PSs within E + P and mill, *p <* 0.05; ^m > n^ comparison between E + Ps within mill and PS, *p <* 0.05; ^y > z^ comparison between PSs within mill, *p <* 0.05; ^Y > Z^ comparison between mills within PS, *p <* 0.05; ^V > W^ comparison between E + Ps within PS, *p <* 0.05; ^v > w^ comparison between PSs within E + P, *p <* 0.05.

**Table 9 animals-12-02707-t009:** Results for 38-day-old broilers showing apparent ileal nutrient digestibility coefficients (DM) of finisher diets processed according to treatments with two mill types at three particle sizes (PS), and submitted to conditioning with or without expander (E + P) prior to pelleting.

Item	Starch	Met	Cys	TSAA	Lys	Thr	Arg	Ile	Leu	Val	His	Phe	Gly	Ser	Pro	Ala	Asp	Glu
Mill type																		
Hammer	0.97	0.94	0.84	0.90	0.89	0.83	0.93	0.89	0.90	0.88	0.90	0.88	0.85	0.88	0.89	0.89	0.88	0.91
Roller	0.97	0.94	0.84	0.89	0.89	0.84	0.93	0.89	0.90	0.88	0.90	0.88	0.85	0.88	0.89	0.89	0.88	0.91
PS, mm																		
0.8	0.97	0.94	0.85	0.90	0.90	0.84	0.93	0.89	0.90	0.88	0.90	0.88	0.85	0.88	0.89	0.89	0.88	0.92
1.2	0.97	0.94	0.84	0.90	0.90	0.84	0.93	0.89	0.90	0.88	0.90	0.88	0.85	0.88	0.89	0.89	0.88	0.91
1.6	0.97	0.93	0.83	0.90	0.89	0.83	0.92	0.89	0.89	0.87	0.89	0.87	0.84	0.87	0.89	0.88	0.87	0.91
E + P																		
With	0.97 ^a^	0.94 ^a^	0.85 ^a^	0.90 ^a^	0.90 ^a^	0.85 ^a^	0.93 ^a^	0.92 ^a^	0.90 ^a^	0.89 ^a^	0.90 ^a^	0.89 ^a^	0.86 ^a^	0.89 ^a^	0.90 ^a^	0.89 ^a^	0.89 ^a^	0.92 ^a^
Without	0.96 ^b^	0.93 ^b^	0.83 ^b^	0.89 ^b^	0.88 ^b^	0.82 ^b^	0.92 ^b^	0.90 ^b^	0.89 ^b^	0.88 ^b^	0.89 ^b^	0.87 ^b^	0.84 ^b^	0.87 ^b^	0.88 ^b^	0.88 ^b^	0.88 ^b^	0.91 ^b^
SEM	0.007	0.015	0.02	0.017	0.02	0.025	0.013	0.02	0.02	0.02	0.016	0.017	0.02	0.02	0.02	0.02	0.02	0.01
*p*-value																		
Mill	0.76	0.61	0.46	0.53	0.98	0.79	0.76	0.88	0.89	0.98	0.82	0.87	0.90	0.89	0.93	0.90	0.96	0.91
PS	0.70	0.50	0.13	0.22	0.42	0.31	0.62	0.49	0.74	0.43	0.42	0.42	0.24	0.34	0.72	0.65	0.32	0.79
E + P	0.001	0.014	0.001	0.001	0.011	0.001	0.002	0.001	0.002	0.001	0.001	0.001	0.001	0.001	0.002	0.004	0.001	0.001
Mill*PS	0.054	0.73	0.19	0.43	0.55	0.25	0.52	0.32	0.24	0.32	0.41	0.26	0.31	0.21	0.39	0.32	0.19	0.21
Mill*E + P	0.23	0.40	0.26	0.34	0.28	0.38	0.36	0.32	0.40	0.34	0.23	0.28	0.27	0.36	0.41	0.35	0.19	0.36
E + P*PS	0.056	0.17	0.08	0.12	0.09	0.12	0.06	0.08	0.09	0.12	0.08	0.056	0.07	0.06	0.06	0.17	0.06	0.06
Mill*PS*E + P	0.85	0.70	0.65	0.71	0.64	0.90	0.80	0.84	0.19	0.83	0.79	0.88	0.85	0.92	0.81	0.83	0.94	0.84

^a > b^ LSmeans in column superscripted with different letters differ significantly according to Tukey–Kramer test, *p <* 0.05; TSAA = total sulfuric amino acids.

**Table 10 animals-12-02707-t010:** Results for 36-day-old broilers showing apparent total-tract nutrient digestibility coefficients from a finisher broiler diet ground using two types of mill at three particle sizes (PS), submitted to conditioning with or without expander (E + P) prior to pelleting.

Item	AME_N_, MJ/kg	OM ^1^	EEh ^2^
Mill type			
Hammer	15.15	0.73	0.78
Roller	15.11	0.73	0.77
PS			
0.8	15.12 ^ab^	0.73 ^ab^	0.79
1.2	15.36 ^a^	0.75 ^a^	0.79
1.6	14.91 ^b^	0.72 ^b^	0.75
E + P			
With	15.39	0.75	0.80
Without	14.87	0.72	0.75
SEM	0.44	0.02	0.03
*p*-value			
Mill	0.730	0.980	0.340
PS	0.005	0.020	<0.001
E + P	<0.001	<0.001	<0.001
Mill*PS	0.310	0.820	0.520
Mill*E + P	0.005	0.001	0.270
E + P*PS	0.440	0.200	0.014
Mill*PS*E + P	0.220	0.430	0.670

^a > b^ LSmeans in column superscripted with different letters differ significantly according to Tukey–Kramer test. (*p <* 0.05); ^1^ OM = organic matter; ^2^ EEh = acid hydrolyzed ether extract.

**Table 11 animals-12-02707-t011:** Breakdown of the significant interactions as presented in Table 10.

Item		AME_N_, MJ/kg	OM	EEh
Mill type	E + P			
Hammer	With	15.25 ^dD^	0.74 ^dD^	
	Without	15.04 ^dD^	0.73 ^dD^	
Roller	With	15.53 ^dD^	0.76 ^dD^	
	Without	14.70 ^eE^	0.71 ^dE^	
E + P	PS, mm			
With	0.8			0.82 ^vV^
	1.2			0.83 ^vV^
	1.6			0.75 ^wV^
Without	0.8			0.76 ^vW^
	1.2			0.76 ^vW^
	1.6			0.74 ^vV^

^d > e^ Comparison between mills within E + P, *p <* 0.05; ^D > E^ comparison between E + Ps within mill, *p <* 0.05; ^V > W^ comparison between E + Ps within PS, *p <* 0.05; ^v > w^ comparison between PSs within E + P, *p <* 0.05.

## Data Availability

No new data were created or analyzed in this study. Data sharing is not applicable to this article.

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
