# Peer review of "Towards Large Particle Size in Compound Feed: Using Expander Conditioning Prior to Pelleting Improves Pellet Quality and Growth Performance of Broilers"

_animals, 2022, doi:10.3390/ani12192707_

Round 1

Reviewer 1 Report

Manuscript

Towards large particle size in broiler compound feed: using expander conditioning prior to pelleting improves pellet quality and growth performance

This is very interesting manuscript on diets processing for poultry. Some suggestions were pointed bellow:

A suggestion to improve title description would be :

Using expander conditioning prior to pelleting improves pellet quality and growth performance of broilers

Simple summary: L14 delete of animal origin

L16- delete On this way

L17 – replace of the diet by in the diet

L19- two different conditionings

L20- replace when they receive diets by when fed diets

L23-24 – suggestion

Delete On the same hand

The use of a roller mill, coarse particles, and expander, resulted in increased energy utilization, nutrient digestibility, and feed conversion ratio of broiler chickens

Abstract

L26 – delete the (before pellets)

L28 – to generate pelleted feeds…. Delete which were fed to broilers

L28 – replace feed processing experiment by arrangement

L29 - 0.8, 1.2 or 1.6 mm)

L30 – replace and fed to by the number of broilers per experimental unit… i.e. with 6 replicated of 12 unsexed Ross 308 broilers each.

L213 – Question? A total of 884 unsexed one-day-old, Ross 308 broiler chickens, vaccinated against Marek, were placed in 36 pens (4 birds/m2 , 12 per pen)

A 2x3x2 factorial arrangement = 12 treatments x 6 rep = 72 experimental units x 12 birds per pen = 864 broilers, is that correct? Please review

L33 – The E+P

Use AMEn instead of AMEN here and in all the text

L41 - P=0.019)

L50 – ; broiler

Introduction

L56 – replace like by as

L114- The objective of this study was

L118 – The responses were

L197- The PDI

L199 – A total of 10 pellets per replicate? Please explain better

Table 1 – please indicate the meaning of the *

L230 – delete degree, same for L236

Tables of results- please use x instead of * to present the interaction in the factorial arrangement

I would be useful if growth performance data and carcass data before metabolizability and digestibility results, because performance was evaluated in the entire period and digestibility can explain performance responses.

Table 5 and 6 – include age of broilers in the title

Very useful manuscript, I recommend acceptance.

Author Response

Animals 1876471 – Towards large particles size in broiler compound feed: using expander conditioning prior to pelleting improves pellet quality and growth performance.

We are very thankful for the reviewers for their valuable contribution to improve the quality of the manuscript. The suggestions for improvement were mostly accepted, and the point-by-point response is below. The manuscript was revised by a native speaker.

OBS: due to the suggestions, the corrected point is not at the same place as in the first submitted version.

Point by point report to reviewers:

Reviewer 1:

A suggestion to improve title description would be: Using expander conditioning prior to pelleting improves pellet quality and growth performance of broilers.

A: Partially accepted and done.

Simple summary

L 14: Delete of animal origin.

A: Done.

L 16: Delete On this way.

A: Done.

L 17: Replace of the diet by in the diet.

A: Done.

L 19: Two different conditioning.

A: Done.

L 20: Replace when they receive diets by when fed diets.

A: Done.

L 23 and 24: Suggestion, delete On the same hand.

A: Done.

L 22: The use of a roller mill, coarse particles, and expander, resulted in increased energy utilization, nutrient digestibility, and feed conversion ratio of broiler chicken.

A: Accepted and done.

Abstract

L 26: Delete the (before pellets).

A: Done.

L 28: To generate pelleted feeds… delete which were fed to broilers.

A: Done.

L 28: Replace feed processing experiment by arrangement.

A: Done.

L 29: 0.8, 1.2 or 1.6 mm).

A: Done.

L 30: Replace and fed to by number of broilers per experimental unit, i.e. with 6 replicated of 12 unsexed Ross 308 broilers each.

A: Done.

L 213: Question? A total of 884 unsexed one-day-old, Ross 308 broiler chickens, vaccinated against Marek, were placed in 36 pens (4 birds/m2, 12 per pen). A 2x3x2 factorial arrangement = 12 treatments x 6 rep = 72 experimental units x 12 birds per pen = 864, is that correct? Please review.

A: Done, the correct number is 864.

L 33: The E+P.

A: Done.

Use AMEn instead of AMEN here and in all the text.

A: Although the literature brings AMEn mostly, the letter N is the symbol for Nitrogen. Considering it we judge more correct use N instead of n.

L 41: P=0.019)

A: Done.

L 50: ; broiler.

A: Done.

Introduction

L 56: Replace like by as.

A: Done.

L 114: The objective of this study was.

A: Done.

L 118: The responses were.

A: Done.

L 197: The PDI.

A: Done.

L 199: A total of 10 pellets per replicate? Please explain better.

A: Done.

Table 1: Please indicate the meaning of the *.

A: Done.

L 230: Delete degree, same for L 236, now 247.

A: Done.

Tables of results: please use x instead of * to present the interaction in the factorial arrangement.

A: Done.

It would be useful if growth performance data and carcass data and carcass data before metabolizability and digestibility results because performance was evaluated in the entire period and digestibility can explain performance responses.

A: Done, it changes the number of the lines, which were corrected to find the places where the corrections were done, considering the suggestions of both reviewers.

Table 5 and 6: Include age of broilers in the title.

A: Done.

Reviewer 2

L 16: Check the wording here.

R: Done.

L 25: What do you mean by animal responses.

R: It means animal performance. Corrected.

L 29: Provide scientific name on first mention here.

R: Done.

L 49: Consider implications and future directions in more detail here.

R: Done

L 50: Some of the keywords are already included in the title. Remove any keywords that are in the title and use new terms to increase paper discoverability.

R: Done.

L 52: Numbers.

R: Done

L 53: Industries.

R: Done.

L 57: Sizes.

R: Done.

L 58: Sizes.

R: Done

L 65: Find a citation to support this.

R: Done.

L 77: There are many grammatical errors, particularly associated with the use of plural versus singular terms. Please review the work thoroughly to reduce these errors.

R: Done.

L 84: Fine what?

R: Done.

L 88: With the pH dropping in the.

A: Done.

L 92: Ground.

A: Done.

L 95: Changes.

A: Done.

L 105 to 107: this sentence doesn’t make sense – can you rephrase?

A: Done.

L 115: The objectives.

A: Done.

L 134: Broiler finisher?

A: Done.

L 214: Shavings.

A: Done.

L 223: What do you mean here? Check meaning of unfastened?

A: Done.

L 290: Did you apply a correction factor for multiple tests?

A: Yes, LSmeans comparison adjusted by Tukey-Kramer test.

L 319: Broiler.

A: Done.

Table 2: Was a correction factor applied? What does bold mean here?

A: Means were adjusted and LSmeans presented. Bolds are used to highlight the significant factors, however we removed it.

L 340: Fine particles?

A: Done.

L 347: Fine particles.

A: Done.

L 387: This doesn’t make sense here.

A: Done.

L 399. Explain the bold here too.

A: bold removed because it was just to highlight P values lower than 0.05.

L 460 to 462: Sentence is quite confused here.

A: Done.

L 464: As the final step.

A: Done.

L 465: Be specific on what response you measured.

A: Done.

L 481: Place this in the results section.

A: Done.

Figure 1: Place this in the results section.

A: Done.

L 595: Cite the source here.

A: Done.

L 639: This doesn’t quite make sense. Can you rephrase?

A: Done.

L 676: Journal names should be consistently italicized – see author guidelines.

A: Done.

L 683. And & required here.

A: Following the author guidelines, just “;” is used between authors names.

L 697: No and required here.

A: Done.

Reviewer 2 Report

Dear Authors,

Thank you for submitting this paper that investigates the use of different particle sizes in the development of broiler diets. This is an interesting paper with some potential value for those involved in agriculture.

At current however, there seem to be some large revisions required in the manuscript to ensure the work is scientifically robust. I have attached the PDF version of the manuscript with specific comments. Additionally, please consider the following points: 

1. Proof reading. Many parts of the work are written with confusing grammar, and the scientific value of the work suffers as a result. I would strongly suggest a full proof read is due, ideally by a native english speaker.

2. Tests correction. The number of tests could have resulted in false positive test results. Please state if any correction factors (e.g. Bonferroni) have been applied. If none have been used, please apply some.

3. Findings. Currently, the findings of the work are not entirely clear. Please evaluate the results carefully to identify the practical implications of this study and what they mean for those working in the industry.

Proof reading. Many parts of the work are written in poor English, and the scientific value of the work suffers as a result. I would strongly suggest a full proof read is due, ideally by a native english speaker.

Author Response

Animals 1876471 – Towards large particles size in broiler compound feed: using expander conditioning prior to pelleting improves pellet quality and growth performance.

We are very thankful for the reviewers for their valuable contribution to improve the quality of the manuscript. The suggestions for improvement were mostly accepted, and the point-by-point response is below. The Englisch language was revised by a native speaker.

OBS: due to the suggestions, the corrected point is not at the same place as in the first submitted version.

Reviewer 2

L 16: Check the wording here.

R: Done.

L 25: What do you mean by animal responses.

R: It means animal performance. Corrected.

L 29: Provide scientific name on first mention here.

R: Done.

L 49: Consider implications and future directions in more detail here.

R: Done

L 50: Some of the keywords are already included in the title. Remove any keywords that are in the title and use new terms to increase paper discoverability.

R: Done.

L 52: Numbers.

R: Done

L 53: Industries.

R: Done.

L 57: Sizes.

R: Done.

L 58: Sizes.

R: Done

L 65: Find a citation to support this.

R: Done.

L 77: There are many grammatical errors, particularly associated with the use of plural versus singular terms. Please review the work thoroughly to reduce these errors.

R: Done.

L 84: Fine what?

R: Done.

L 88: With the pH dropping in the.

A: Done.

L 92: Ground.

A: Done.

L 95: Changes.

A: Done.

L 105 to 107: this sentence doesn’t make sense – can you rephrase?

A: Done.

L 115: The objectives.

A: Done.

L 134: Broiler finisher?

A: Done.

L 214: Shavings.

A: Done.

L 223: What do you mean here? Check meaning of unfastened?

A: Done.

L 290: Did you apply a correction factor for multiple tests?

A: Yes, LSmeans comparison adjusted by Tukey-Kramer test.

L 319: Broiler.

A: Done.

Table 2: Was a correction factor applied? What does bold mean here?

A: Means were adjusted and LSmeans presented. Bolds are used to highlight the significant factors, however we removed it.

L 340: Fine particles?

A: Done.

L 347: Fine particles.

A: Done.

L 387: This doesn’t make sense here.

A: Done.

L 399. Explain the bold here too.

A: bold removed because it was just to highlight P values lower than 0.05.

L 460 to 462: Sentence is quite confused here.

A: Done.

L 464: As the final step.

A: Done.

L 465: Be specific on what response you measured.

A: Done.

L 481: Place this in the results section.

A: Done.

Figure 1: Place this in the results section.

A: Done.

L 595: Cite the source here.

A: Done.

L 639: This doesn’t quite make sense. Can you rephrase?

A: Done.

L 676: Journal names should be consistently italicized – see author guidelines.

A: Done.

L 683. And & required here.

A: Following the author guidelines, just “;” is used between authors names.

L 697: No and required here.

A: Done.

Round 2

Reviewer 2 Report

Dear Authors,

Many thanks for submitting this revised version of the manuscript for review. You have taken into account the feedback provided on the initial review of the paper.

In light of the revisions, the paper is now in a much better position for consideration.